# Exploring a New Natural Treating Agent for Primary Hypertension: Recent Findings and Forthcoming Perspectives

**DOI:** 10.3390/jcm8112003

**Published:** 2019-11-16

**Authors:** Shian-Ren Lin, Shiuan-Yea Lin, Ching-Cheng Chen, Yaw-Syan Fu, Ching-Feng Weng

**Affiliations:** 1Department of Life Science and Institute of Biotechnology, National Dong Hwa University, Hualien 97401, Taiwan; d9813003@gms.ndhu.edu.tw (S.-R.L.); abc57142001@yahoo.com.tw (C.-C.C.); 2Graduate Institute of Cancer Biology and Drug Discovery, Taipei Medical University, Taipei 110, Taiwan; 3Department of Anatomy, Kaohsiung Medical University, Kaohsiung 80708, Taiwan; shiuanyea@yahoo.com.tw; 4Camillian Saint Mary’s Hospital Luodong,160 Zhongzheng S. Rd. Luodong, Yilan 26546, Taiwan; 5Department of Biomedical Science and Environmental Biology, Kaohsiung Medical University, Kaohsiung 80708, Taiwan; 6Center for Infectious Disease and Cancer Research, Kaohsiung Medical University, Kaohsiung 80708, Taiwan; 7Department of Basic Medical Science, Center for Transitional Medicine, Xiamen Medical College, Xiamen 361023, China

**Keywords:** Hypertension, herbal, natural compounds, RAAS, molecular docking

## Abstract

Primary hypertension describes abnormally-high systolic/diastolic blood pressure in a resting condition caused by various genetic or environmental risk factors. Remarkably, severe complications, such as ischemic cardiovascular disease, stroke, and chronic renal disease have led to primary hypertension becoming a huge burden for almost one-third of the total population. Medication is the major regimen for treating primary hypertension; however, recent medications may have adverse effects that attenuate energy levels. Hence, the search for new hypotensive agents from folk or traditional medicine may be fruitful in the discovery and development of new drugs. This review assembles recent findings for natural antihypertensive agents, extracts, or decoctions published in PubMed, and provides insights into the search for new hypotensive compounds based on blood-pressure regulating mechanisms, including the renin-angiotensin-aldosterone system and the sympathetic/adrenergic receptor/calcium channel system.

## 1. Introduction

Hypertension is a homeostatic blood pressure shift from normal (<120/80 mmHg) to abnormal levels (>140/90 mmHg) which is categorized into the following four stages: normal pressure (<120/80 mmHg), elevated pressure (120–139/80–89 mmHg), stage 1 hypertension (140–159/90–99 mmHg), and stage 2 hypertension (>160/100 mmHg) [1,2,3]. Only 5–10% of hypertensive patients can be diagnosed from certain causes e.g., aldosteronism, gestation, or renal artery stenosis (secondary hypertension); however, the other 90–95% of hypertensive patients have unknown or multiple reasons for their condition [4,5]. Despite uncertain causes, some predisposing factors for primary hypertension, such as genetic variation, obesity, improper eating habits including excessive sodium and alcohol consumption, tobacco and marijuana use, and physical unfitness, are summarized [6,7,8].

About 31.1% of adults were hypertensive worldwide in 2010, with 28.5% of them living in high-income countries [9]. Notably, 32.5% of adults (17.3% of total citizens) are in various stages of hypertension in China [10]. Furthermore, the overall mortality of high systolic blood pressure is 20,526/100,000 people, and the age-standardized disability-adjusted life year rate (DALY) is 242.5/100,000 people [11,12]. These harsh statistics remind us that hypertension is highly prevalent, and is a burden on society in terms of the associated health expenditure. Generally, hypertensive patients endure various symptoms that are difficult to self-assess, such as headaches, dizziness, shortness of breath, chest pain, palpitations, and nosebleeds. However, common complications of hypertension are more severe, e.g., stroke, ischemic cardiovascular disease (ICD), and chronic kidney disease (CKD) (see Table 1) [13,14,15]. Therefore, the treatment and medical care of hypertension is a critical issue for elevating the well-being of mankind.

The clinical foci of hypertension treatment comprise ameliorating blood pressure and preventing complications. Generally speaking, by using a blood pressure lowering approach, medication is the first choice, while the management life style gives constructive support in enhancing adherence [3,16]. Numerous agents, e.g., verapamil, amlodipine, and captopril, have been applied in the treatment of hypertension; the antihypertensive effects of these drugs have been well established [3,17]. Beyond medication, patients and clinicians propose a variety of supports from pharmaceuticals, society, family, and patient’s personal pharmacogenomics for managing hypertension. Social isolation, low income, depression, or psychological distress have been ascertained as causes of withdrawal during treatment [3,16,18,19]. Statistical analyses from the investigation of African American women, Latino migrants, seasonal farmworkers, Chinese people, and people from south-west Nigeria have illustrated that social supports could promote patient adherence to treatment via sharing information and assisting in bringing about life style changes [20,21,22,23]. Surprisingly, about 10.3% of hypertension patients develop hypertensive resistance during the treatment regimen, which may be typified by a lower response to medications [24]. Nevertheless, several adverse effects—e.g., hyponatremia [25], hyperkalemia [26], and edema [27]—of recent medications do not facilitate the medical compliance of patients [28]. Accordingly, finding new blood pressure lowering agents with medical benefits for hypertensive patients, and to promoting the progress of new drug discovery, are increasingly important. Expectedly, cumulative research evidence has illustrated that natural compounds or herbal medicines are believed to be equipotent with fewer adverse effects compared to synthetic drugs [29]. In this review, we summarize recent findings pertaining to hypertension remedies from natural products and research approaches according the causes and shedding mechanism of hypertension that could provide new insights into perspective studies for the research and discovery of natural antihypertensive agents.

## 2. Physiological Regulation Mechanism in Blood-Pressure

Basically, the body undertakes the regulation of blood-pressure via two strategies: total blood volume and blood flow. Total blood volume is mainly mediated by renin-angiotensin-aldosterone systems (RAAS), while blood flow is predominantly regulated by neurogenic hypertensive system and nitric oxide (NO) [30,31]. Consequently, current hypertensive medication also targets the aforementioned mechanisms (RAAS and neurogenic hypertensive system) to alleviate high blood pressure [1]. The following section will individually address each blood-pressure-controlling mechanism to summarize our present knowledge of the regulation of blood pressure, medications, and screening tools for natural compounds.

### 2.1. Renin-Angiotensin-Aldosterone System (RAAS)

#### 2.1.1. Mechanical Action of RAAS in Blood-Pressure Controlling

RAAS is long-term regulator of blood pressure composed of renin (an aspartic protease produced by the juxtaglomerular apparatus of kidney), angiotensin converting enzyme (ACE, a zinc-dependent dicarboxyprotease secreted from the lungs), angiotensinogen (a hormone precursor), and aldosterone (mineralocorticoid hormone from adrenal cortex, the endpoint effector). Angiotensinogen is converted into angiotensin II (Ang II) because it is sequentially cleaved by renin and ACE; subsequently, Ang II stimulates the adrenal gland cortex to produce aldosterone [32]. Aldosterone raises blood-pressure via two phases: (i) potentiating Na^+^/K^+^-ATPase activity on sodium and water reabsorption of renal tubular cells; and (ii) consequently managing (i.e., increasing or decreasing) blood volume (see Figure 1A), as well as vascular smooth muscle cells for vasoconstriction [33,34]. To understand the cause of elevating blood pressure, the consequence of a decrease of sympathetic input and an increase in parasympathetic input to the heart is observed. Second, if the baroreceptors detect that blood pressure is too high, the cardio-regulatory center of the medulla will also decrease the sympathetic input to the blood vessels. This causes vasodilation instead of vasoconstriction, which decreases the total peripheral resistance and concurrently decreases blood pressure (see Figure 1B). In this section, we will not discuss the sympathetic nerve regulating system; rather, we will mainly focus on the regulation of the RAAS system in order to present a comprehensive review of potential targets for lowering blood pressure.

#### 2.1.2. Conventional RAAS Modulators in Hypertensive Medication

According to the physiological mechanism of RAAS, several targets have been identified: ACE, the angiotensin II type-1 receptor (AGTR1), the aldosterone receptor, and Na^+^/K^+^-ATPase. ACE inhibitors (e.g., Captopril, Ramipril, and Fosinopril) and Ang II receptor blockers (e.g., Valsartan, Irbesartan, Candesartan, and Olmesartan) directly suppress Ang II and aldosterone production in the adrenal gland [35,36]. Aldosterone receptor inhibitors (also known as K^+^-sparing diuretics, e.g., Spironolactone, Pironolactone, and Eplerenone) allosterically inhibit the interactions of aldosterone with its receptor. Additionally, Na^+^/K^+^-ATPase blockers (thiazides such as benzylhydrochlorothiazide and indapamide) lead to reduced Na^+^/K^+^-ATPase activity and reduced water reabsorption [37]. Moreover, RAAS modulators can improve hypertension through directly diminishing the total blood volume. Hence, RAAS modulators play a major role in primary hypertension treatments [3]. However, RAAS modulators tend to preserve K^+^ and eliminate Na^+^, which would increase the risks of edema, hyperkalemia, and hyponatremia [25,26,37]. This adverse effect needs to be taken into account when using these modulators.

#### 2.1.3. Methods for Screening RAAS Modulators

Numerous available models are used in the search for new RAAS modulators through in silico, in vitro, and in vivo approaches. Chemical induction, renal artery ligation, and genetic modification are common methods with which to generate the in vivo hypertensive model [38]. Of note, chemical induction is conducted by administrating 11-deoxycorticosterone acetate (DOCA) or Ang II, which results in transient hypertension [39,40]. Furthermore, transient hypertension transforms into persistent hypertension during chemical induction, which occurs when combined with a high-salt diet [40]. In general, the intravenous (i.v) or subcutaneous (s.c) administration of Ang II is a common model [41,42]. Ang II infusion could keep high levels of Ang II in serum; however, in experiments, animals needed to be anesthetized for more than 7 days [41,42]. Therefore, this technique is rarely used in drug screening but the risk of hypertensive complications, e.g., aneurysm, nephropathy, preeclampsia, and cognitive impairment [42,43,44,45]. Renal artery ligation is the most reported animal model for simulating reno-vascular hypertension, the most common cause of secondary hypertension [46]. Renal artery stenosis can be simulated by partial ligation on the left side or on both sides of the renal artery, leading to reno-vascular hypertension in animal models [47]. The genetic hypertension model is particular to spontaneous hypertension and includes over half of the hypertension population [38]. A well-known spontaneous hypertensive rodent model mainly used in the study of RAAS-induced hypertension is the spontaneous hypertensive rat (SHR) [48]. SHR is originally derived from inbreeding the Wistar–Kyoto (WKY) rats with hypertension, and is suitable for studying either hypertension or renal disease and left ventricular diastolic dysfunction [49,50]. Despite spontaneous hypertension, another primary cause of primary hypertension is obesity, which accounts for 60–75% of primary hypertension cases [51]. To study obesity-induced hypertension, the New Zealand Obese strain (NZO) is appropriate because of its spontaneously polygenic obesity and insulin resistance [52]. In diet-induced hypertension, a high-fat and high-salt diet could be alternatively applied to obtain the desired model [53]. A high-fat diet is another choice for obesity-induced hypertension mice model, and is usually coupled with type-2 diabetes [54]. When mice are fed with a high-salt diet for 4–8 weeks, glucocorticoid resistance could be found in animals, eventually resulting in hypercortisolemia, which is an appropriate model for the diuretic screening of hypernatremia, a cause of hypertension [55,56]. Currently, newly-bred, spontaneous Phase-1 hypertension (P1-HT) rats were successfully generated and used to confirm *Plantago asiatica* seed extract as a potentially antihypertensive herb when compared with SHR [57].

ACE inhibitors can also be tested by in silico to in vitro modeling. In in silico modeling, compound modules and ACE structural modules are employed to simulate ligand/receptor interactions. Higher molecular interaction strengths could demonstrate a greater possibility of being potent inhibitors [58]. By comparing interacting patterns, different ligand types might interact with different amino acid residues in ACE. For small molecules, Gln^281^, Lys^511^, and Tyr^520^ of ACE are essential for binding, forming a hydrogen bond with the ligand [59]. In the search of ACE peptide inhibitors, Val^518^, His^383^, His^387^, and His^51^, are the most important, forming electrostatic and hydrophobic interactions with peptide inhibitors and Zn^2+^ [60]. In the enzymatic approach, ACE activity could be determined by interacting with peptide derivatives that are composed of peptides and colorimetric indicators. The colorimetric assays for ACE are very sensitive, fast, and easy to use [61]. However, in vitro enzymatic conditions will not ever be equivalent to in vivo conditions, due to the highly complex composition of body fluids [62]. The foreseeable gap between enzymatic screening and in vivo or in vitro analyses needs more studies to shorten the distance.

Other RAAS targets, including AGTR1, aldosterone receptor, and Na^+^/K^+^-ATPase, could not be tested by colorimetric or chemical assay. Recently, cell membrane chromatography has used cell membranes as a stationary phase, allowing the molecules to interact with receptors on the cell membrane [63]. This technique could theoretically overcome the limitation of enzymatic assays for which this principle is not suitable for evaluations of ligand/receptor interactions [64]. Nonetheless, the interaction between cell membranes and ligand candidates could not distinguished whether they were activators or inhibitors. On the other hand, human kidney proximal tubule epithelial cells HK-2 have been identified as carriers of mineral corticoid receptor, angiotensin receptor type 2, and Na^+^/K^+^ ATPase expression [65,66,67]. Moreover, aldosterone is shown to be an apoptotic inducer in proximal tubule cells [68]. This evidence shows that HK-2 could be an ideal cellular platform for the screening of diuretics, be it Na^+^/K^+^ ATPase inhibitor or aldosterone receptor inhibitor. Hou et al. reported on the anti-hypertensive efficacy of malate and aspartate via NO and L-arginine generation in HK-2 cells [69]. Using HK-2 cells, they demonstrated that interleukin-17A raises blood pressure through upregulating Na^+^/Cl^−^ cotransporter and Na^+^/H^+^ exchanger 3, which are also the main characters of sodium reabsorption in the proximal and distal convoluted tubules [70]. These studies have confirmed that HK-2 is a promising model for the screening of anti-hypertensive agents.

#### 2.1.4. RAAS Modulators Identified from Nature

Numerous RAAS modulators have been identified in nature that can be classified as ACE and AGT1R inhibitors (Table 2). The identified ACE inhibitors include polyphenolics, alkaloids, and short-chain polypeptides (Table 2). Particularly in chalcones, these widespread flavonoids serve as calcium flux or ACE inhibitors [71,72]. Some Traditional Chinese medicines such as Dohaekseunggi-tang and Huatuo exhibit anti-hypertensive activity in benchside experiments, and are worthy of further investigation (Table 2). Some proteins digested and isolated from bonito, milk, sardines, seaweeds, or royal jelly show antihypertensive activity via inhibiting ACE activity. These bioactive peptides are currently available commercially [73,74].

To screen AGT1R inhibitors, two parallel approaches, i.e., directly blocking receptor activity and reducing receptor expression, will be performed. Table 2 shows natural compounds that could block AGT1R activity. As far as we know, only five plants, i.e., *Alisma orientale* Juzepczuk, *Cynodon dactylon*, *Eucommia ulmoides* Oliver, *Astragalus membranaceus*, and *Salvia miltiorrhizae*, have traditionally been used as herbal medicines and contain identified natural AGT1R inhibitors (see Table 2). Two traditional Chinese medicines, i.e., the injection of DanHong and Liuwei Dihuang taken orally have also been proven to inhibit AGT1R activity in an in vitro assay (Table 2). On the other hand, Yiqi Huaju formula and Songling xuemaikang capsules have been found to attenuate hypertension in vivo through down-regulating AGT1R expression (Table 2). Tanshinone IIA, a flavonoid from *S. miltiorrhizae*, directly blocks AGT1R activity and reduces AGT1R expression via TGF-1β/Smads signaling *pathway* in vivo (Table 2). These reports provide the potential for insights into finding new RAAS modulators from natural sources.

Despite the data presented in Table 2, several common foods have also shown antihypertensive potency in multiple phases. Probiotics can diminish high blood pressure by mediating immunomodulation and adjusting sympathetic nerve activity [125]. Bioactive components of avocado (*Persea americana*) contain alkaloids, unsaturated fatty acids, and flavonoids that lessen either high arterial pressure or body weight [126]. Tomato (*Solanum lycopersicum*), a common vegetable, is another example of a natural ACE inhibitor. During the past two years, five clinical trials studying the health impact of tomato extracts on a high cardiovascular risk population pointed to a reduced risk of hypertension or other cardiovascular diseases [127,128,129,130,131]. By exploring the detailed mechanisms, the major bioactive component of tomato, lycopene, was identified as an ACE inhibitor and antioxidant that might be the main substance responsible for its antihypertensive efficacy [132]. Additionally, coffee and tea are well known as antihypertensive foods [133]. Tea and coffee have antihypertensive potential through diuresis [134,135] and changes in emotional states [136,137]. The main components of tea and coffee [138,139], caffeine, theophylline, chlorogenic acid, and epigallocatechin, are known as antihypertensive compounds through their ability to modulate salt reabsorption and immunomodulation [137,139,140,141,142]. However, the antihypertensive efficacy of coffee and tea has been challenged by the latest epidemiological studies in different regions and on both genders. According to a statistical analysis of US adults and a meta-analysis from PubMed, coffee intake reduces the likelihood of hypertension [143,144]. In Singapore, antihypertensive potential versus daily coffee intake exhibits a U shape, and tea consumption slightly increases the risk of hypertension occurrence [145]. Through further investigation by gender, three epidemiological studies and meta-analyses reveal that only women could alleviate hypertensive risk through the consumption of coffee, except in the case of postmenopausal women [146,147,148]. These controversial results remind us of the discrepancy between recent knowledge and our common sense concerning coffee/tea consumption and lowering hypertensive risk.

### 2.2. Neurogenic Hypertensive System

#### 2.2.1. Action Mechanism of Neurogenic Hypertensive System in Blood Pressure Managing

Unlike RAAS, which regulates blood pressure by controlling total blood volume, sympathetic nerves govern blood pressure by adjusting heart rate and vascular diameter. Sympathetic nerves send neurotransmitters (mainly epinephrine and norepinephrine) as signals to the sinus node and smooth muscle cells within blood vessels [149]. Once target cells receive signals via binding to adrenergic receptors (ARs), intracellular Ca^2+^ flow forms via the L-type Ca^2+^ channel, causing smooth muscle cell contraction [149]. ARs are divided into two types: α-type and β-type, for which α-AR mainly induces smooth muscle contraction and β-AR excites the sinus node and relaxes smooth muscle cells (see Figure 1B). This balancing pathway plays a critical role in mediating blood pressure.

#### 2.2.2. Conventional Drugs for Modulating the Neurogenic Hypertensive System

The main actors for the neurogenic hypertensive system are ARs and the Ca^2+^ channel, so major modulators in this system are predominantly focused on ARs and Ca^2+^ channel. α-blockers (e.g., Doxazosin and Terazosin) block signals that are sympathetic toward α-AR, which induces vasocontraction [150]. β-blockers (e.g., Propranolol and Bisprolol) can hinder heart rates [151]. Critically, calcium blockers (e.g., Nifedipine and Diltiazem) act at the end of the neurogenic hypertensive system by inhibiting both heart rate and vasoconstriction [152]. Ordinarily, sympathetic hyperactivity is an important cause of primary hypertension in youth [153,154]. Hence, AR and calcium blockers are the first line of hypertension medication, especially primary hypertension in the young- or middle-aged population [155,156]. However, a rapid decrease of blood pressure might sometimes trigger ischemic heart disease [152]. Also, ARs are reported to induce sexual dysfunction like erectile dysfunction, ejaculatory disorder, and reduced sexual desire [157,158]. These adverse effects might diminish willingness to follow medical advice.

#### 2.2.3. Approaches to Screening Controllers for Neurogenic Hypertensive System

The screening methods for controllers of neurogenic hypertensive systems are mainly in vitro, ex vivo, and in vivo models. In vitro assays are performed only at the cellular or tissue levels due to the absence of enzymes in the neurogenic hypertensive system. Chinese hamster ovary cells (CHO), vascular smooth muscle cells (VSMC, isolated from thoracic aorta or the commercial cell line, A7r5), and the cardiomyocyte cell line h9c2 are used to evaluate the overall inhibiting activity of AR or calcium. CHO cell derivatives like CHO-K1 and CHO-S are used as carriers for expressing α-AR or β-AR and for measuring the downstream cAMP change after the induction of the neurotransmitter [159,160]. These cell models exert a searching approach for a potential ligand. VSMC directly responds to signals from norepinephrine or external calcium supplements [121,161]. β-ARs and the Ca^2+^ channel have demonstrated their expression in the h9c2 cell line [162,163]. Hence, the h9c2 cell has been applied in the study of pathological or cardioprotective efficacy in ischemic heart disease or cardiac hypertrophy for years [164,165]. However, the application of h9c2 is not widely used in screening β-blockers or calcium blockers in drug development; this weakness needs further exploration. An ex vivo model usually comprises 4 mm lengths of rat aortic rings isolated from the thoracic arteries and immersed in Krebs solution for treatment [166]. This explant model could directly observe the diameter changes of the aortic rings after treatment with a neurotransmitter or Ca^2+^ [167].

The two most frequently-used animals in in vivo models for screening neurogenic hypertensive systems are environmental stimulation and spontaneous hypertensive mice [168]. The principle of environmental induction involves using environmental stimulus to trigger psycho-emotional stress, and subsequently arterial hypertension, through sympathetic nerve hyperactivation [169]. Common environmental stimuli include flashing lights, loud noises, restraints, cages, and high/low temperatures [170]. Therefore, continuous stimulation is also useful for investigating neurogenic hypertensive systems. Inherited hypertensive mice, BPH/2J (Schlager), are derived from the hypertensive parental strain BPH/1J, whose hypertensive symptoms are milder than those of BPH/2J [171]. BPH/2J exhibits spontaneously-sympathetic hyperactivity and low RAAS activity that results in a high heart rate and high blood pressure [172,173]. Due to these different causes of hypertension, BPH/2J and SHR represent completely different approaches.

Despite the existence of the wet-lab model, one in silico model is also provided for neurogenic hypertensive systems, i.e., β_2_-AR [174]. By evaluating ligand-binding properties, the essential ligand binding sites of β_2_-AR are identified, i.e., Asp^113^ and Asn^312^ [175]. Also, polar amino acid residues in transmembrane domain 5 of β_2_-AR, and especially Ser^203^ and Ser^207^, play an important role in agonist-binding stabilization [175].

#### 2.2.4. Natural ARs and Calcium Blockers

Compared with AGT1R, blocking the AR and L-type calcium channel is more common for maintaining normal-tension. The AR and calcium channel can directly control blood vessel diameter and vasoconstriction; this activation lessens the blood vessel diameter and causes an increase in blood pressure [152]. Comparing adrenergic blockers and calcium blockers, calcium blockers also affect the heart rate; thereby, calcium blockers are used in arrythmia more frequently than in hypertension [152]. For the treatment of hypertension, Table 3 shows known β-blockers and calcium blockers from natural sources. For α-blockers, tempol (a food in South-East Asia made from fermented soybean) and fermented *Lactobacillus rhamnosus* GR-1 could treat preeclampsia and ameliorate cardiac hypertrophy by inhibiting α-AR [176,177]. These two studies have demonstrated the anti-hypertensive potential of probiotics instead of RAAS. In a review about the pathogenetic characteristics of gut microbiota in hypertension published by Kang and Cai, a possible underlying mechanism affects hypertensive progress through three phases: inflammatory regulation, ACE inhibition, and sympathetic activity adjustment [178]; this area needs further investigation. Again, three coumarins (paeruptorin A, B, and C) isolated from *Peucedanum praeruptorum* Dunn and one alkaloid ((+)-nantenine) from *Nandina domestica* have also been shown to have α-AR blocker activities in vitro and in vivo [179]. These results further revealed that finding adrenergic blockers and calcium blockers from natural sources is a promising avenue of research in new drug discovery and development.

## 3. Conclusions and Remarks

In this article, we have collected recent research and developments of antihypertensive agents from natural sources according to the two main blood-pressure-regulating mechanisms: RAAS and the neurogenic hypertensive system. The implementation of American Heart Association (AHA)/European Society of Cardiology (ESC) guidelines for the management of hypertensive patients, as well as the public health and cost implications, need to be reflected upon [17]. However, drug resistance and poor adherence are urgent issues associated with the treatment of hypertension; these could be a niche for drug development, especially from natural sources [204]. Additionally, folk medicines, medicinal plants, and herbs may be incorporated into the regimen of anti-hypertension treatments according to their experimental efficacy in preclinical trials. Figure 2 summarizes such substances. To accelerate the screening rates, molecular docking is a potential tool for use before any wet-lab validation, including enzymatic tests or cellular assays. However, the absence of target protein modules is an emergent application for molecular docking in antihypertensive compound discovery. Also, recently-developed cellular or enzymatic analyzing methods are not particularly well-suited for the in vivo or in silico discovery, validation, or evaluation of the regulating mechanisms of blood pressure. Nevertheless, numerous compounds or decoctions have been found to have hypotensive efficacy. Moreover, some studies have found other enzymes that could also become antihypertensive targets, such as endothelial NO synthase (eNOS) [123,205,206,207,208,209,210], endothelin-1 receptor [211], and G-protein-coupled receptor kinase 4 (GRK4) [212].

Additionally, the characteristics of the gut–brain axis in blood pressure regulation has received much attention in recent decades. To our knowledge, gut microbiota and the brain could cross talk through various routes, e.g., small microbial metabolites such as sodium butyrate and vitamin D, autonomic nerves system, and intestinal immunocytes [213,214,215,216,217]. A comparison of gut microbiota between healthy and hypertensive people reveals compositional differences, particularly in Firmicutes and Bacteroidetes; this difference might be attributed to the cause of hypertension, at least partially [218]. Accordingly, some studies have focused on the treatment of fluctuating gut microbiota. Probiotic products, such as kefir and lactulose, have been shown to negate hypertensive symptoms in a rat model [125,219,220]. This evidence also offers an insights into the potential of applying natural-source gut-microbiota modulators or traditional Chinese herbal medicine in hypertensive regimens [221]. The exploration of new targets for drug screening to accelerate anti-hypertension drug discovery and development can be more profitable and more effective than other recent methods.

## Figures and Tables

**Figure 1 jcm-08-02003-f001:**
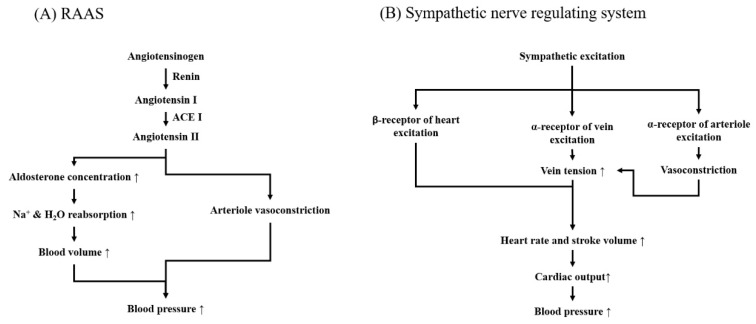
Regulating mechanism for blood pressure. (**A**) RASS; (**B**) Sympathetic nerve regulating system. RAAS: renin-angiotensin-aldosterone systems; ACE I: angiotensin converting enzyme inhibitor.

**Figure 2 jcm-08-02003-f002:**
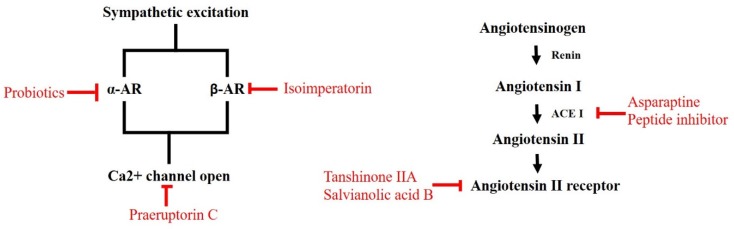
Natural compounds act on blood pressure control.

**Table 1 jcm-08-02003-t001:** Complications of hypertension.

Affect Organ	Complications
Cardiovascular system	Cardiac hypertrophy, heart failure, angina pectoris, coronary heart disease, myocardial infarction.
Artery	Atherosclerosis, aneurysms.
Brain	Stroke (ischemic or hemorrhagic), hypertensive encephalopathy, cognitive decline, dementia.
Eye	Retinopathy.
Kidney	Hypertensive nephropathy, chronic kidney disease.

**Table 2 jcm-08-02003-t002:** RAAS modulators from natural sources.

Source	Compound/Extractions	Evaluated Method	Reference
**Crude extract/decoction for modulating ACE**			
Dohaekseunggi-tang		in vivo	[75]
Huatuo reconstruction pill		in silico	[76]
*Nigella sativa*		enzymatic	[77]
*Cymodocea Nodosa*		+	[78]
*Plantago major*	Seeds extract	+	[79]
*Mucuna pruriens*		+	[80]
*Syzygium cumini*	Seeds extract	in vitro	[81]
*Fucus spiralis*		enzymatic	[82]
*Eucalyptus camaldulensis* *Litsea glaucescens*	Ferment	+	[83]
*Lens culinaris*	Sprouted extract	+	[84]
*Pleurotus eryngii*	polysaccharide	+	[85]
*Phaseolus vulgaris*	Ferment	+	[86]
*Prunus amygdalus* *Pistacia vera*	juice byproduct	+	[87]
Boza	protein extract	+	[88]
*Nitraria sibirica*	alkaloids	in vivo	[89]
*Amaranthus dubius* *Amaranthus hybridus* *Asystasia gangetica* *Galinsoga parviflora* *Justicia flava* *Oxygonum sinuatum* *Physalis viscosa*		enzymatic	[90]
*Tulbaghia violacea*		in vivo	[90]
*Bulbus Fritillaria*		in vivo	[91]
**Crude extracts or decoction for AGT1R**			
*Apocynum venetum* leaf extract		in vitro	[92]
DanHong injection		*+*	[93]
Liuwei Dihuang formula		*+*	[94]
Yiqi Huaju formula		in vivo	[95]
Songling xuemaikang capsule		*+*	[96]
**ACE inhibitors**			
*Ampelopsis Brevipedunculata*	(+)-Hopeaphenol; (+)-Vitisin A	enzymatic	[97]
*Asparagus officinalis*	Asparaptine	+	[98]
*Avena sativa*	WWK, WCY, FLLA	+	[99]
*Bothrops jararaca*	EWPRQIPP, EARPPHPPIPP, EWGRPPGPPIPP, EGGWPRPGP(Glu)IPP,	in vitro	[100]
*Bothrops moojeni*	(Pyro)EKWPPGGKVPP, (Pyro)EKPRPGPEIPP, (Pyro)ENWPWPGPEIPP	enzymatic	[101]
*Cajanus cajan*	VVSLSIPR	+	[102]
*Camellia sinensis*	Epigallocatechin gallate	+	[59]
Chinese Herb	Pentagalloylglucose, Isochlorogenic acid B, Methyl 3,4-di-O-caffeoylquinate, (−)-Epigallocatechin gallate, Epigallocatechin-3-O-Methylgallate	in vivo	[103]
*Cleistanthus collinus*	Cleistanthins A, B	in silico	[104]
*Clerodendron trichotomum*	Acteoside, Isomartynoside, Leucosceptoside A, Martynoside	enzymatic	[105]
*Coptis chinensis*	Berberine	in vitro	[91]
*Coix larchryma-jobi*	GAAGGAF, NPATY	in vivo	[60,106]
*Delphinium sp.*	Cyanin, Delphinidin	enzymatic	[107]
*Desmodium styracifolium*	Carlinoside, Schaftoside, Vicenin 1–3	+	[108]
*Dioscorea opposita* Thunb.	Diascorin	+	[109]
Egg York	Lapslpgkpkpd	+	[110]
*Eucommia ulmoides*	Megastigmane Enantiomers	in silico	[111]
*Glycyrrhiza glabra*	Licochalcone A	enzymatic	[59]
*Glycyrrhiza uralensis*	Echinatin	+	[59]
*Limonium michelsonii*	Isolates	+	[112]
*Mucuna Pruriens*	Genistein	+	[80]
Multisource	Caffeic Acid, Caffeoyl Acetate, Chlorogenic Acid, Ferulic Acid	+	[113]
*Salvia hispanica L.*	LIVSPLAGRL	+	[114]
*Salvia miltiorrhizae*	Salvianolic Acid B	+	[115,116]
	Lithospermic Acid B	+	[105]
*Sargassum wightii*	O-Heterocyclic Analogues	in silico	[117]
*Tamarix hohenackeri*	Chrysoeriol, Quercetin, Isoferulic acid, Methyl-4-O-methylgallate, Gallic acid, Methyl gallate	enzymatic	[118]
*Toona sinensis*	Quercetin, Resveratrol	+	[107]
*Vigna radiata*	KDYRL, VTPALR, KLPAGTLF	+	[119]
Xestospongia Cf. Vansoesti	Salsolinol	+	[115]
**AGT1R inhibitors**			
*Alisma orientale*	23-O-acetylalisol B, Alismol, Alisols A, Alisols B	enzymatic	[120]
*Astragalus membranaceus*	Astragaloside IV	in vitro	[121]
*Cynodon dactylon*	Linoleoylchloride, Diazoprogesteron, Didodecyl Phthalate	enzymatic	[122]
*Eucommia ulmoides*	Megastigmane Enantiomers	in silico	[111]
*Salvia miltiorrhizae*	Salvianolic acid B	in vivo	[123]
*+*	Tanshinone IIA	in vivo	[124]

+, as above; WWK, (Trp)_2_-Lys; WCY, Trp-Cys-Tyr; FLLA, Phe-(Leu)_2_-Ala EWPRQIPP, Glu-Trp-Pro-Arg-Pro-Gln-Ile-(Pro)_2_; EARPPHPPIPP, Glu-Ala-Arg-(Pro)_2_-His-(Pro)_2_-Ile-(Pro)_2_; EWGRPPGPPIPP, Glu-Trp-Gly-Arg-(Pro)_2_-Gly-(Pro)_2_-Ile-(Pro)_2_; EGGWPRPGP(Glu)IPP, Glu-(Gly)_2_-Trp-Pro-Arg-Pro-Gly-Pro-GluIle-(Pro)_2_; (Pyro)EKWPPGGKVPP, PyroGlu-Lys-Trp-(Pro)_2_-(Gly)_2_-Lys-Val-(Pro)_2_; (Pyro)EKPRPGPEIPP, PyroGlu-Lys-Pro-Arg-Pro-Gly-Pro-Glu-Ile-(Pro)_2_; (Pyro)ENWPWPGPEIPP, PyroGlu-Asn-(Trp-Pro)_2_-Gly-Pro-Glu-Ile-(Pro)_2_; VVSLSIPR, (Val)_2_-Ser-Leu-Ser-Ile-Pro-Arg; GAAGGAF, Gly-(Ala)_2_-(Gly)_2_-Ala-Phe; NPATY, Asn-Pro-Ala-Thr-Tyr; LIVSPLAGRL, Leu-Ile-Val-Ser-Pro-Leu-Ala-Gly-Arg-Leu; KDYRL, Lys-Asp-Tyr-Arg-Leu; VTPALR, Val-Thr-Pro-Ala-Leu-Arg; KLPAGTLF, Lys-Leu-Pro-Ala-Gly-Thr-Leu-Phe.

**Table 3 jcm-08-02003-t003:** Natural source β blockers and Ca^2+^ blockers.

Source	Compounds	Test Method	Reference
**Crude extract/decoction**			
β-adrenergic receptor inhibitor			
Banxia Baizhu Tianma Tang		in vitro	[180]
*Paeoniae Rubra*		in vitro	[181,182]
Rhubarbs		in silico	[183]
*Suaeda asparagoides*		in vivo	[184]
Ca^2+^ channel inhibitor			
*Alternanthera sessilis*		in vitro	[185]
*Capparis aphylla*		in vivo	[186]
*Coreopsis tinctoria*		in vitro	[187]
*Gentiana floribunda*		in vivo	[188]
*Jasonia glutinosa*		in vitro	[161]
*Juniperus excelsa*		in vitro	[189]
*Perovskia abrotanoides* essential oil		in vitro	[190]
Pumpkin seed oil		in vivo	[191]
*Ranunculus japoniucus*		in vivo	[192]
*Tiangou Jiangya*		in vitro	[193]
*Viola odorata*		in vivo	[194]
**Pure compounds**			
β-adrenergic receptor inhibitor			
*Nelumbo nucifera*	Higenamine 4′-O-β-d-glucoside	in vitro	[195]
*Notopterygium incisum*	Isoimperatorin	enzymatic	[159]
*Pimpinella anisum*	Trans-anethole	in silico	[196]
Ca^2+^ channel inhibitor			
*Agelanthus dodoneifolius*	Dodoneine	in vitro	[197]
*Glycosmis petelotii*	N-demethylglypetelotine, Glypetelotine	in vitro	[198]
*Peucedanum praeruptorum*	Praeruptorin C	in vivo	[199]
*Polygonum multiflorum*	Emodin	in vitro	[200]
*Sargassum siliquastrum*	Sargachromenol D	in vitro	[201]
*Stephaniae tetrandrae*	Tetrandrine, Fangchinoline	in vivo	[202]
*Stevia rebaudiana*	Stevioside	in vivo	[203]

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
