# Peer review of "Exploring a New Natural Treating Agent for Primary Hypertension: Recent Findings and Forthcoming Perspectives"

_jcm, 2019, doi:10.3390/jcm8112003_

Round 1

Reviewer 1 Report

Quite interesting study concerning natural/herbal agents in hypertension treatment.

However:

1) please try to use more medical language, in table 1 Authors write: "hardening of the arteries"? = do they mean atherosclerosis? or "cerebrovascular plugs" = atherosclerosis, "stroke" = ischemic stroke, hemorrhagic is mentioned as cerebral hemorrhage; "uremia" = better end stage kidney failure' kidney failure itself ia a part of kidney disease and can be omitted in this manner;

2) when you mentioned the examples of drugs from ACE inhibitors or angiotensin inhibitors please use e.g. (line 100), it looks like there are no other representatives from this classes.

3) Amiloride (line 103) is claimed to be a potassium sparing diuretic, but not from the aldosterone receptor antagonists, but sodium channel blocker (despite that channel is regulated by aldosterone) -

Vidt DG. Mechanism of action, pharmacokinetics, adverse effects, and therapeutic uses of amiloride hydrochloride, a new potassium-sparing diuretic. Pharmacotherapy. 1981 Nov-Dec;1(3):179-87, please correct that.

4) Figure 1 has poor resolution, please try to improve it.

Author Response

1) please try to use more medical language, in table 1 Authors write: "hardening of the arteries"? = do they mean atherosclerosis? or "cerebrovascular plugs" = atherosclerosis, "stroke" = ischemic stroke, hemorrhagic is mentioned as cerebral hemorrhage; "uremia" = better end stage kidney failure' kidney failure itself is a part of kidney disease and can be omitted in this manner;

A: We appreciate reviewer’s valuable suggestion. We have replaced all suggested terms in Table 1 as medical terms.

2) when you mentioned the examples of drugs from ACE inhibitors or angiotensin inhibitors please use e.g. (line 100), it looks like there are no other representatives from this classes.

A: Thank for reviewer’s suggestion. We have added ”e.g.” before commercial drugs.

3) Amiloride (line 103) is claimed to be a potassium sparing diuretic, but not from the aldosterone receptor antagonists, but sodium channel blocker (despite that channel is regulated by aldosterone) -Vidt DG. Mechanism of action, pharmacokinetics, adverse effects, and therapeutic uses of amiloride hydrochloride, a new potassium-sparing diuretic. Pharmacotherapy. 1981 Nov-Dec;1(3):179-87, please correct that.

A: We thank for reviewer’s correction. Amiloride has been removed.

4) Figure 1 has poor resolution, please try to improve it.

A: Thank for reviewer’s comment. The quality of Figure has been modified.

Reviewer 2 Report

Review on the mansucript "Exploring a New Natural Treating Agent for Primary Hypertension: Recent Findings and Impending Perspectives"

The authors provide a review on the current literature concerning traditional and natural treatment options in hypertension. This is on the one hand´s side an interesting approach. But it has also a dangerous component: High blood pressure is an important cardiovascular risk factor! And we have very good medication to treat high blood pressure. It is nice if there are supportive treatment options – but in fact it must be made clear that patients with hypertension HAVE TO be treated in the generally accepted medical way: By ACE inhibitors and betablockers etc. This has to be made clear and thus the manuscript should be substancially overworked!

Figure 1 is not a nice figure.

Figures are lacking to make it more concrete.

The models for studying hypertension are not well explained: Mouse models and cell culture models have to be better explained – figure schemes? The Angiotensin II mouse model is missing completely which ist he most important one. Examples often seem to be picked out arbitrarily.

The relevant experts on the field gut microbiota and hypertension are not cited.

All in all, much of the article seems to be touched superficially and needs more profound analysis.

And: What is the intention and the real goal of the review? It should become more critical. Then it could become a good review that helps to find a way to support current antihypertensive treatment options with traditional / natural compounds.

Author Response

The authors provide a review on the current literature concerning traditional and natural treatment options in hypertension. This is on the one hand´s side an interesting approach. But it has also a dangerous component: High blood pressure is an important cardiovascular risk factor! And we have very good medication to treat high blood pressure. It is nice if there are supportive treatment options – but in fact it must be made clear that patients with hypertension HAVE TO be treated in the generally accepted medical way: By ACE inhibitors and beta-blockers etc. This has to be made clear and thus the manuscript should be substantially overworked!

Figure 1 is not a nice figure.

A: Thank for reviewer’s comment. The quality of figure has been modified.

Figures are lacking to make it more concrete.

A: We thank the reviewer’s suggestion and we do agree this comment, we have added more concrete description of Figure 1 in L93-102.

The models for studying hypertension are not well explained: Mouse models and cell culture models have to be better explained – figure schemes? The Angiotensin II mouse model is missing completely which is the most important one. Examples often seem to be picked out arbitrarily.

A: We appreciate reviewer’s comment and we have added description of angiotensin II infusion model in L127-132.

The relevant experts on the field gut microbiota and hypertension are not cited.

A: Thank for reviewer’s valuable comment. We have discussed potential of gut-brain axis in hypertension regulation in section of future remarks at L331-344.

All in all, much of the article seems to be touched superficially and needs more profound analysis.

And: What is the intention and the real goal of the review? It should become more critical. Then it could become a good review that helps to find a way to support current antihypertensive treatment options with traditional / natural compounds.

A: We appreciate with reviewer’s suggestion. We have modified aims and made it more concise in L69-72.

Round 2

Reviewer 2 Report

The paper is still missing this point:

It must be made clear that patients with hypertension HAVE TO be treated in the generally accepted medical way: By ACE inhibitors and betablockers and so on. The general guidelines muts be cited (AHA ESC antihyertensive treatment). This has to be made clear right in the beginning and also in the end. We already have good tretment options. Sure, herbal medication can be used on top or for developing new options - but this here MUST be included!

Author Response

The paper is still missing this point:

It must be made clear that patients with hypertension HAVE TO be treated in the generally accepted medical way: By ACE inhibitors and beta-blockers and so on. The general guidelines must be cited (AHA ESC antihypertensive treatment). This has to be made clear right in the beginning and also in the end. We already have good treatment options. Sure, herbal medication can be used on top or for developing new options - but this here MUST be included!

A: We appreciate reviewer’s valuable suggestion. We have added the new description at L64-66 and 327-335.